# *Genista tridentata* Phytochemical Characterization and Biological Activities: A Systematic Review

**DOI:** 10.3390/biology12111387

**Published:** 2023-10-30

**Authors:** Inês Martins Laranjeira, Alberto Carlos Pires Dias, Filipa Lacerda Pinto-Ribeiro

**Affiliations:** 1Life and Health Sciences Research Institute (ICVS), School of Medicine, Campus of Gualtar, University of Minho, 4710-057 Braga, Portugal; inesmartinslaranjeira@gmail.com; 2ICVS/3B’s—PT Government Associate Laboratory, 4806-909 Guimarães, Portugal; 3CITAB—Centre for the Research and Technology of Agro-Environmental and Biological Sciences, University of Trás-os-Montes e Alto Douro, 5000-801 Vila Real, Portugal; 4Centre of Molecular and Environmental Biology (CBMA), University of Minho, Campus de Gualtar, 4710-057 Braga, Portugal

**Keywords:** traditional medicine, biological properties, *Genista tridentata*, *Pterospartum tridentatum*, nutraceuticals, phytochemicals

## Abstract

**Simple Summary:**

This study systematically reviewed the literature on the bioactivities and phytochemical profile of *Genista tridentata* (L.) Willk, which is an edible plant used in folk medicine. Four databases (PubMed, GoogleScholar, Repositórios Cientificos de Acesso Aberto de Portugal (RCAAP), and ScienceDirect) were searched from inception up to 31 December 2022. Among 34 eligible papers, the flowers and aerial parts were extensively studied, with aqueous extracts being the most commonly used. The findings suggest that *Genista tridentata* has high potential as a natural source of antioxidants and preservatives for the food/cosmetic industry, offering health benefits. Additionally, its year-round collectability provides an advantage for industrial applications.

**Abstract:**

*Genista tridentata* (L.) Willk., known as “prickled broom”, is a Leguminosae (Fabaceae) species native to the Iberian Peninsula, Morocco, Algeria, and Tunisia. It is used in folk medicine as an anti-inflammatory, for gastrointestinal and respiratory disorders, rheumatism, and headaches, to lower blood pressure, against hypercholesterolemia and hyperglycemia. This study aimed to systematically review the literature on the bioactivities and phytochemical profile of *Genista tridentata* to understand its pharmacological potential. For this, four electronic databases (PubMed, GoogleScholar, Repositórios Cientificos de Acesso Aberto de Portugal (RCCAP), and ScienceDirect) were searched from inception up to 31 December 2022. From a total of 264 potentially eligible studies considered for screening, 34 papers were considered eligible for this systematic review. The sampling included 71 extracts, collected mainly in Portugal. *Genista tridentata* extracts present a high level of flavonoids and phenolic compounds. The flowers and aerial parts of the plant were the most studied, and aqueous extracts were the most used. The results predict a high potential for the application of *Genista tridentata* as a new source of natural antioxidants and preservatives for the food industry with subsequent health benefits, such as the production of nutraceuticals. Moreover, the results indicate that the plant can be collected at all seasons of the year, which represents a benefit for the industry.

## 1. Introduction

Wild edible plants are an important piece of the cultural and genetic heritage of distinct world regions, representing high importance, predominantly in rural and suburban areas [1]. Furthermore, they are interesting sources of bioactive compounds and need recognition as considerable contributors to human health promotion and disease prevention [2].

*Genista tridentata* (L.) Willk. (the recognized name for this species), also known as *Pterospartum tridentatum* (L.) Willk. (the commonly used name in both scientific literature and commercially available extracts). Among other synonyms, *Chamaespartum tridentatum* (P.) Gibbs is also used [3,4]). Commonly known as “prickled broom”, it is a Leguminosae (Fabaceae) species belonging to the subfamily Papilionoideae [5,6]. In line with scientific literature and the Global Biodiversity Information Facility database [7], the recorded countries of origin for the plant remain consistent, comprising Portugal, Spain, and Morocco. However, it is important to mention that the Plants of the World Online (POWO) database [8] also lists Algeria and Tunisia as potential countries of origin for this plant. This shrub can be found in the understory of *Arbutus unedo*, *Pinus*, and *Eucalyptus* forests, as well as in abandoned lands. It grows spontaneously up to 100 cm in acidic soils [9] and presents yellow flowers with a typical odor in alternate branches and coriaceous winged stems [10]. Traditionally, it is harvested in the spring between March and June.

*G. tridentata* are an aromatic plant that is very important in Portuguese gastronomy. The leaves are conventionally used as a condiment/spice for the seasoning of traditional rice and meat dishes [11]. Moreover, fresh or shade-dried flowers of *G. tridentata* are also used in folk medicine, in infusions, decoctions, and tonics [12] as anti-inflammatory [13,14,15], diuretic and depurative of the liver [5,11,16,17]. It is commonly used to ameliorate colds [5,18], in digestive disorders [5,18,19,20], intestinal [21,22] and urologic problems [5,11,15,16,18], and rheumatism [5,11,16]. Additionally, it is also used for respiratory disorders [5,6,13,15,18,23], headaches [5], to lower blood pressure [5,6,18], against hypercholesterolemia [5,6,18,20,22] and hyperglycemia [5,6,11,16,17,18,23,24], and in weight loss programs [5].

This study aimed to systematically review the literature on the biological activities of *G. tridentata* extracts. It also focused on its phytochemical profile that could be relevant for understanding its use in folk medicine and its pharmacology potential, highlighting this nutraceutical potential.

## 2. Materials and Methods

This systematic review was carried out according to the PRISMA guidelines (Preferred Reporting Items for Systematic Reviews and Meta-Analyses) [25].

### 2.1. Search Methods for Identification of Studies

#### 2.1.1. Electronic Searches

For the systematic review, 4 electronic databases (PubMed, Google Scholar, RCAAP, and ScienceDirect) were searched from inception up to 31 December 2022. The keywords used for database search were: “*Pterospartum tridentatum*” OR “*Genista tridentata*” OR “*Chamaeospartium tridentatum*”.

#### 2.1.2. Searching Other Resources

The authors did not proceed to the identification of studies through the screening of citations, nor were the authors contacted.

### 2.2. Criteria for Considering Studies for This Review

#### 2.2.1. Types of Studies

For the evaluation of the biological activities of *G. tridentata*, we included only original research articles published in peer-reviewed journals.

#### 2.2.2. Inclusion and Exclusion Criteria

The inclusion criteria were (i) original research articles, (ii) reports providing quantitative data, and (iii) reports focusing on the evaluation of the biological activities and phytochemical profile of *G. tridentata* extracts or essential oils. The exclusion criteria were (i) review articles, (ii) grey literature, (iii) reports written in a language other than English, and (iv) reports with commercial pure compounds not directly derived from plant biomass.

#### 2.2.3. Outcomes

The primary outcomes were synthesizing data concerning the biological activities of *G. tridentata* and its phytochemicals profiles.

### 2.3. Data Collection and Analysis

#### 2.3.1. Selection of Studies

After searching the database and removing duplicates, two independent researchers (I.M.L. and F.P.R.) screened the titles to select relevant papers, followed by screening the abstracts to evaluate their content. Any disagreements were resolved by consensus between the researchers.

#### 2.3.2. Data Extraction and Management

Two authors (F.P.R. and I.M.L.) extracted data independently into an Excel file. Any discrepancies were resolved by discussion or, if necessary, by the third author (A.D.).

### 2.4. Quality Assessment

The quality assessment of the included articles was performed using the ARRIVE GUIDELINES for animal research: Reporting In Vivo Experiments [26], whenever applicable, as no specific guideline is in use for plant-based studies. Accordingly, we adjusted the guidelines for plant research by including two additional parameters, namely sample characterization and sample extraction.

## 3. Results

### 3.1. Literature Search

A schematic representation of the literature screening process is provided in Figure 1. In the first triage, after the removal of the duplicates, 2001 results were identified. From these, 719 were excluded because of the language criteria, then 1282 were excluded as their topic was outside the scope of this review. The remaining 264 articles were screened by title and abstract, with 230 being excluded for the following reasons: 88 because of document type (review articles and grey literature), 79 did not provide data concerning biological or pharmacological activities; 1 was excluded because it reported mixed results (*G. tridentata* extract was mixed with other plant species); 13 were excluded because they reported results of isolated compounds not directly related to *G. tridentata* and, 14 were excluded because of missing data. Thus, 34 articles were included in the systematic review.

The reports included identification and analyses of the phytochemical compounds present in *G. tridentata* extracts and their bioactivities.

### 3.2. Extraction Procedures, Plant Parts Used, and Sampling

Most extracts used in the studies were derived from aqueous extraction (61.8%) [5,6,9,11,13,15,16,17,18,19,20,22,24,27,28,29,30,31,32,33,34,35] followed by methanolic (20.6%) [14,17,23,36,37,38,39], ethanolic (14.7%) [4,28,35,40] and acetone (2.9%) [41] extracts. One work used hydroglycolic extraction [42]. One work did not disclose the type of solvent used to obtain the extract [43].

Considering the part of the plant used for extraction, 52.9% used flowers [5,9,11,12,13,15,20,23,24,27,29,32,34,35,36,37,39,42,43], 20.6% aerial parts [4,11,16,28,33,41], 14.7% leaves [13,19,38,40,41,42], 5.9% stems and leaves [11,23], 2.9% stems [5], and 2.9% leaves and flowers [6]. One work used in vitro culture (2.9%) [24], and four works (11.8%) did not disclose the part of the plants used to obtain the extracts [17,18,22,30].

In terms of sampling location, 47.1% were collected in Portugal [4,5,9,11,12,15,16,17,23,24,27,28,35,37,38,39,40], 11.8% were collected in Spain [14,40,41,42], and 38.2% were obtained from herbal shops [6,13,19,20,22,29,30,31,32,33,34,36,43]. One work (2.9%) [18] did not disclose how extracts were obtained.

Additionally, 35.3% of the samples were collected during the flowering period [4,5,11,12,15,16,24,32,36,41,42] and 8.8% during the dormancy period [17,20,24]. The remaining 61.8% of the works do not disclose this information [6,9,13,14,17,18,19,22,23,27,28,29,30,31,33,34,35,37,38,39,40].

### 3.3. Phytochemical Characterization

The main compounds found are flavonoids, as well as hydroxycinnamic acids and hydroxybenzoic acids (Table 1, Figure 2). Additionally, extracts collected in the flowering period (May), as well as flowers, presented a more diverse phytochemical profile than extracts collected during the rest of the year.

Other classes of compounds found in the literature include essential oils (EOs) (Table 2). Grosso et al. (2007) [11] studied the EOs isolated by hydrodistillation and distillation-extraction of flowers, stems and leaves, and aerial parts of different populations. Another study [29] also characterized EOs of *G. tridentata* using the same methodology. The results show that *G. tridentata* samples presented a yellowish oil with a yield of <0.05% (*v*/*w*). The dominant components of the oils were phenylpropanoids, more abundant in aerial parts, and oxygen-containing monoterpenes in the flowers, stems, and leaves. Additionally, Faria et al. (2013, 2016) [31,33] reported cis-theaspirane and trans-theaspirane as the main components.

### 3.4. Biological Activities

Considering the bioactivities, 44.1% of the works are related to antioxidant activity [5,6,9,12,13,15,19,20,23,24,28,30,32,34,35,38], followed by antifungal (17.6%) [17,34,38,39] and antibacterial (11.8%) [34,37], anti-inflammatory [4,14,15,34,40], and nematocidal [29,31,33] activities (14.7 and 8.8%, respectively). The cytotoxicity of the extract was evaluated in five works [6,15,20,23,34] (14.7%). Only two works studied the antihyperglycemic activity (5.9%) [6,27]. Other activities studied included anti-acetylcholinesterase [20] and anti-cholesterol [22] activity (2.9% each).

#### 3.4.1. Antioxidant Activity

The antioxidant activity of *G. tridentata* was evaluated by numerous assays, from which the DPPH (2,2′-diphenyl-1-picrylhydrazyl radical) scavenging assay and TBARS (thiobarbituric acid reactive substances) were the most used ones (Table 3).

Gião et al. (2007) [13] used the ABTS (2,2′-azinobis-(3-ethylbenzothiazoline-6-sulfonic acid)) test to evaluate the antioxidant activity, showing boiled infusions (leaves) exhibit the highest values (0.260 ± 0.030 gL^−1^ of ascorbic acid equivalent), in comparison to cold infusions (Powder: 0.164 ± 0.036 gL^−1^ of gallic acid equivalent; Leaves: 0.057 ± 0.025 gL^−1^ of gallic acid equivalent).

Three works refer to the antioxidant activity index, showing a high antioxidant activity of *G. tridentata* methanolic extracts of flowers (1.7 ± 0.06%) and moderate antioxidant activity of steam and leaves (0.7 ± 0.06%) [23]. Other studies identified a strong antioxidant activity of aqueous extracts (1.30 ± 0.05% [28]; 106.6 ± 0.7 µmol g^−1^ dw [30]); however, they did not disclose the part of the plant used to obtain the extract.

Gonçalves et al. (2020) [35] showed *G. tridentata* extracts were able to inhibit NO and O^2−^ in a concentration-dependent manner (infusion extract: NO: IC_50_ = 95.41 ± 0.96 μg/mL; O^2−^ IC_50_ = 23.31 ± 2.82 μg/mL; hydroethanolic extract: NO: IC_50_ = 281 ± 2.65μg/mL; O^2−^ IC_50_ = 26.76 ± 1.83 μg/mL).

When Pinela et al. (2011) [12] and Roriz et al. (2014) [43] evaluated the antioxidant activity of *G. tridentata* methanolic extracts of flowers by *β*-*carotene bleaching* assay, they obtained different results depending on the processing method (120 min: Freeze-drying: 0.14 ± 0.02 mg/mL; Shade-drying: 0.13 ± 0.01 mg/mL; methanolic maceration: 0.48 ± 0.09 mg/mL). Likewise, Ferreira et al. (2012) [6] also showed antioxidant activity of aqueous extracts of leaves and flowers (30 min: 101.8 + 10.7%AA; 60 min: 169.5 + 17.2%AA).

Pinela et al. (2011) [12] and Roriz et al. (2014) [43] evaluated the reducing power of *G. tridentata* methanolic extracts of flowers to reduce Fe (III) to Fe (II) and obtained similar results (freeze-drying: 0.13 ± 0.00 mg/mL; shade-drying: 0.13 ± 0.00 mg/mL; methanolic maceration: 0.11 ± 0.00 mg/mL), independently of the extraction method used.

One study [16] assessed the iron chelating activity of *G. tridentata* aqueous extracts of aerial parts, demonstrating that the hot infusions were the most effective although there was not a dose-dependent correlation between the concentration of the extract and the degree of inhibition (Cold: 0.4 mg/mL ext-1:94.55 ± 4.20%; 0.8 mg/mL ext-1:84.32 ± 1.17%; 1.6 mg/mL ext-1:68.32 ± 4.59%; Hot: 0.4 mg/mL ext-1:67.24 ± 2.34%; 0.8 mg/mL ext-1:63.48 ± 1.69%; 1.6 mg/mL ext-1:52.94 ± 0.62%).

Vitor et al. (2004) [9] demonstrated that 3-*O*-glucoside isoquercitrin and prunetin, isolated from an aqueous extract of *G. tridentata*, were shown to prevent endothelial oxidative damage and have radical scavenging activity inhibiting O^2−^ generation.

Caleja et al. (2019) [34] and Gonçalves et al. (2020) [35] evaluated the inhibition of free radical-induced membrane damage in erythrocytes, showing that *G. tridentata* aqueous extracts of flowers were able to protect the erythrocytes population for 120 min (60 m: 37.7 ± 0.9 μg/mL; 120 m: 69 ± 2 μg/mL) [34] and both, aqueous and hydroethanolic extracts of flowers, prevented hemolysis in a concentration-dependent manner (aqueous extract: IC_50_ = 21.73 ± 0.95 μg/mL; hydroethanolic extract: IC_50_ = 28.43 ± 2.26 μg/mL) [15,36]. Additionally, flower extracts prevented the oxidation of hemoglobin in a concentration-dependent manner, with the results demonstrating that the aqueous extract was the most efficient (IC_50_ = 52.87 ± 2.22 μg/mL), followed by the hydroethanolic extract (IC_50_ = 54.03 ± 6.15 μg/mL) [35].

Moreover, in two studies [19,35], aqueous extracts of *G. tridentata* were assayed for their capacity to protect deoxyribose from degradation. Results showed that the strongest antioxidant effect was associated with hot water extraction, compared to cold extraction.

#### 3.4.2. Anti-Inflammatory Activity

Bremner et al. (2009) [14] assessed the anti-inflammatory activity of methanolic, petroleum ether, and ethyl acetate extracts of *G. tridentata* using the TNF-α inhibition assay in human monocytes. The results indicated an inhibition level between 80 and 60% for all extracts. Also, more recently, Simões et al. (2020) [4] evaluated the anti-inflammatory activity of *G. tridentata* ethanolic extracts (roots, stems, and leaves) through the inhibition of LPS-NO (lipopolysaccharide-nitric oxid) production and demonstrated a downregulation of the Nos2 gene. The other two studies also demonstrated a decrease in the transcription of the pro-inflammatory genes IL1b, IL6, and Ptgs2 with stems and root extracts. LPS-NO production was decreased in flower-based *G. tridentata* aqueous extract [15,34] and in ethanolic extract [15]. Mota et al. (2022) [40] evaluated the myeloperoxidase (MPO) inhibition by ethanolic extract of *G. tridentata* and demonstrated an IC_50_ of 0.032 ± 0.004 mg/mL.

#### 3.4.3. Antifungal and Antibacterial Activity

The results of the antifungal activity of *G. tridentata* are presented in Table 4. The results showed that *G. tridentata* extracts are effective against six *Candida* strains, namely *C. albicans*, *C. glabrata*, and *C. parapsilosis* species, while the majority of *C. tropicalis* strains were resistant [37]. The antifungal activity of aqueous extract of *G. tridentata* flowers was also demonstrated against *Aspergillus niger*, *Aspergillus versicolor*, *Penicillium funiculosum*, *Penicillium aurantriogriseum* and *Penicillium verrucosum* [15,34].

Table 4 also shows the results for the screening of the antibacterial activity of *G. tridentata* against *S. aureus* and *E. coli*, *Salmonella Typhimurium*, *Bacillus cereus*, and *Listeria monocytogenes*. *G. tridentata* extracts display moderate to strong antibacterial activity against *S. aureus* [17].

#### 3.4.4. Cytotoxicity

The cytotoxicity of *G. tridentata* aqueous extract was assessed using a human colorectal epithelial adenocarcinoma cell line (Caco-2) and a human cervical adenocarcinoma cell line (HeLa) [20]. Additionally, the cytotoxicity of *G. tridentata* flower aqueous extract was also assessed using a human hepatocarcinoma cell line (HePG2) [6]. The results showed no toxicity in the cell lines evaluated.

Ferreira et al. (2012) [6] also evaluated the toxicological effects of *G. tridentata* flower aqueous extract in mitochondrial respiratory rates (state 4 and state 3 respiration and FCCP-stimulated respiration) and respiratory indexes (respiratory control ratio and P/O ratios) in rat liver mitochondria. The results demonstrated no toxicity of the extracts since no identifiable interactions with respiratory enzymes were detected.

Caleja et al. (2019) [34] and Garcia-Oliveira et al. (2022) [15] also assessed the cytotoxicity of aqueous and ethanolic extracts of flowers using four human tumor cell lines: HeLa (cervical carcinoma), HepG2 (hepatocellular carcinoma), MCF-7 (breast adenocarcinoma), and NCI-H460 (non-small cell lung cancer), as well as a non-tumor cell primary culture PLP2 (porcine liver). The results indicate no cytotoxicity for non-tumor cells and for NCI-H460 and MCF-7 cells. Also, they verified an inhibition of 50% of cell growth in the HeLa and HepG2 cell lines (251 ± 6 µg/mL and 262 ± 11 µg/mL, respectively). Garcia-Oliveira et al. (2022) [15] demonstrated an inhibition of 50% of cell growth in aqueous and ethanolic extract in the NCI-H460 (GI:142.7 ± 5.3 µg/mL and 160.5 ± 5.3 µg/mL, respectively), HeLa (GI:83.2 ± 6.5 µg/mL and 102.9 ± 10.6 µg/mL, respectively), MCF-7 (GI:129.1 ± 6.3 µg/mL and 146.8 ± 6.5 µg/mL, respectively), and HepG2 (GI:123.1 ± 19.1 µg/mL and 132.4 ± 8.5 µg/mL, respectively) cell lines.

#### 3.4.5. Nematocidal/Nematotoxicity Activity

Barbosa et al. (2010) [29] and Faria et al. (2013, 2016) [31,33] evaluated the nematocidal activity of EOs of *G. tridentata*, highlighting a moderate to strong activity. One report showed a mortality rate of 100% against *Bursaphelenchus xylophilus* exposed for 24 h to a 2 mg/mL of *G. tridentata* flower-based EO [29]. Moreover, Faria et al. (2013) [31] demonstrated a moderate (80–61%) nematotoxicity activity against *Bursaphelenchus xylophilus* when exposed to 2 µL/mL of *G. tridentata* EO. Likewise, Faria et al. (2016) [33] demonstrated inhibition of *Meloidogyne chitwoodi* eggs hatching rates (>90%) when exposed to 2 µL/mL^−1^ EO.

#### 3.4.6. Other Activities

Gonçalves et al. (2020) [35] evaluated the in vitro capability of flower-based *G. tridentata* extracts to inhibit α-glucosidase. Aqueous and hydroethanolic extracts inhibited α-glucosidase in a concentration-dependent manner (IC_50_ = 130 ± 0.90 μg/mL; IC_50_ = 148 ± 2.54 μg/mL, respectively).

Currently, there is just one scientific report [27] regarding the antihyperglycemic properties of *G. tridentata* in vivo. The antihyperglycemic effect of the aqueous extract was evaluated in normoglycemic rats using an oral glucose tolerance test. Rats were administered the *G. tridentata* extract (300 mg/kg). The extract exhibited a significant antihyperglycemic effect in the initial 30 min after the glucose challenge, although, at later time points, the blood glucose levels increased, and the extract demonstrated a paradoxical hyperglycemic effect. The same authors also analyzed blood glucose in normoglycemic rats after administration of sissotrin and isoquercitrin, demonstrating an opposite effect in glucose tolerance: isoquercitrin (100 mg/kg) significantly decreased blood glucose levels for 30 min while sissotrin (100 mg/kg) produced a significant increase in blood glucose levels for 60 min.

Acetylcholinesterase (AChE) enzymatic activity of *G. tridentata* flowers aqueous extracts was evaluated by Serralheiro et al. (2013) [20]. The results showed IC_50_ values for AChE activity of 1090 ± 4 µg/mL. After 4 h digestion, the remaining enzymatic activities for the gastric and pancreatic juices were 105.8 ± 5.5% and 103.5 ± 17.3%, respectively.

Falé et al. (2014) [22] assessed the molecular mechanism of cholesterol reduction by *G. tridentata* aqueous extracts demonstrating a plasma total cholesterol reduction of 22% in vivo (humans; 10 g/L of dry infusion on the day of the experiment). This work also included in vitro studies to evaluate the inhibitory concentration of HMG-CoA (IC_50_ = 329.04 ± 21.24 μg/mL) and the permeation in Caco-2 monolayers cells where they verified the bioavailability of some bioactive compounds in basolateral and intracellular compartments (isoquercitin—basolateral: 5.09 ± 0.54 and intracellular: 5.64 ± 0.31%; biochanin A—basolateral: 6.30 ± 0.52 and intracellular: 3.69 ± 0.06%), as well as the apparent permeability coefficient (Papp) (isoquercitin—13.09 ± 1.39 × 10^−7^ cm/s; biochanin A—16.20 ± 1.33 × 10^−7^ cm/s).

### 3.5. Quality Assessment

The quality assessment of the works was performed using the ARRIVE quality guidelines, with each criterion being rated with a score between 0 and 1 (0—Absent; 0.5—Incomplete or not applicable; 1—Complete). About 56.25% of the studies were considered good quality, scoring higher than 15 out of 20, and 43.75% were considered moderate quality, rating between 12 and 15 out of 20. The mean score was determined for each work, and the total mean score for the quality of included works was 15.18 ± 1.14 (Table 5).

## 4. Discussion

### 4.1. Phytochemical Characterization

The phytochemical profile (Table 1) clearly demonstrates that extract composition varies considerably with the extraction method, probably due to the solubility of the compounds. Methanolic and ethanolic solvents are less polar in nature, disrupting cell walls (nonpolar structures) and triggering the release of phenolic compounds; while water presents a high polarity index, it is best to extract compounds with higher polarity [35]. As the extraction solvent influences the plant extract composition [37,39], reports should carefully describe the extraction method to facilitate comparison between laboratories. The location and the vegetative stage also influence the phytochemical profile of extracts; consequently, it is important to be careful in the analyses of data from the works that do not indicate the sampling period. Plant extracts collected in the flowering period (May), as well as flowers, displayed a more diverse phytochemical profile than extracts collected during the rest of the year.

Furthermore, the in vitro extracts exhibited a different composition content than wild extracts [24]. These variations could be explained because some compounds are not essential in in vitro plants as a defense mechanism against harmful environmental conditions, and consequently, they do not produce them [24].

Moreover, works performing a phytochemical characterization of *G. tridentata* are scarce, especially when using other extraction solvents, such as acetone. Another important parameter is the quantification of the phytochemical compounds identified, often overlooked by researchers.

Regarding EOs (Table 2), the variability between samples from the same location and year indicates that the chemical composition of *G. tridentata* oils is not a consequence of climatic factors in distinct years but a consequence of other intrinsic factors related to the plant (ratio plant/organ, vegetative state, genetics) and its interaction with the environment (type of the soil, climate, harvest time during the day) [44].

### 4.2. Biological Activities

#### 4.2.1. Plant Bioactive Compounds as Antioxidants

There is no general procedure that can determine the antioxidant capacity of all samples precisely and quantitatively [45]. Standardized approaches to evaluate antioxidant activity should follow specific requirements, and consequently, the procedures fall into two broad categories: radical scavenging activity and lipid peroxidation inhibition [46].

The variability of assays facilitates the establishment of *G. tridentata* potential [3]. However, it is impossible to compare between essays since there is no standardization of procedures, and *G. tridentata* extracts are obtained with different solvents and from different locations. Additionally, not all evaluated works performed replicates of the antioxidant tests [5,12,18,24,27,41], so they should be repeated.

Furthermore, the anti-oxidative effects of plant extracts on deoxyribose and DNA were promising concerning practical applications of *G. tridentata* as an ingredient in the formulation of nutraceutical beverages and foods, as well as cosmetic formulations, since they add a protective effect.

The phytochemical characterization (Table 1 and Table 3) of *G. tridentata* extracts demonstrates a high level of phenolic compounds, especially flavonoids. Structurally, phenolic compounds possess an aromatic ring with one or more hydroxyl substitutes. The aromatic feature and highly conjugated system with multiple hydroxyl groups make these compounds good electron/hydrogen atom donors, neutralizing free radicals and other ROS [47].

The antioxidant activity of phenolics can be based on hydrogen atom transfer and/or single-electron transfer. Nevertheless, the antioxidant potential of a phenolic compound depends on the number and the position of hydroxyl groups in the molecule [47,48].

Likewise, the degree of hydroxylation also influences the antioxidant activity. The longer distance separating the carbonyl group and the aromatic ring appears to improve the antioxidant capability. Additionally, a greater number of hydroxyl aromatic rings, such as in flavonoids, increased the antioxidant activity [47].

Importantly, flavonoids and their glycosylated derivates in *G. tridentata* extracts were stable during in vitro digestion, which indicates a strong antioxidant activity [20]. Nevertheless, this analysis should be repeated since this single work evaluated gastric and pancreatic digestion, and their quality assessment indicates some deficiencies (Table 5).

Some studies showed a significant synergistic effect between bioactive compounds and, consequently, an increase in antioxidant activity [49,50]. Therefore, the high antioxidant capacity of *G. tridentata* extracts is probably due to a synergistic interaction between their phytochemical components rather than due to a single chemical element.

#### 4.2.2. Anti-Inflammatory Activity

Several health benefits attributed in traditional medicine to *G. tridentata* infusions or decoctions are linked to their anti-inflammatory activity [5,51]. Nevertheless, scientific reports focused on the anti-inflammatory potential of *G. tridentata* extracts are scarce and do not identify the mechanism of action. The phytochemical characterization of *G. tridentata* extracts demonstrated their richness in flavonoids, frequently correlated with anti-inflammatory activity [3,52]. Additionally, numerous studies demonstrated the anti-inflammatory potential of biochanin A [53,54,55,56], prunetin [57,58], genistein [59,60], rutin [61,62] and taxifolin [63,64] in decreasing inflammatory mediators as TNF-α, inhibiting the production of pro-inflammatory cytokines (IL-1β, IL-6, IL-33) and of pro-inflammatory enzymes as nitric oxide synthases. Pinto et al. (2020) [3] provided a comprehensive review that explores the potential mechanisms of action of several bioactive compounds also found in *G. tridentata* concerning their anti-inflammatory activity.

However, it would be important to evaluate the synergistic effects of the bioactive compounds, as well as the evaluation of extracts obtained with different solvents. Importantly, the in vivo evaluation of *G. tridentata* extracts could be central to understanding the mechanisms of action and establishing the pharmacological effects of the plant’s extracts.

#### 4.2.3. Antifungal and Antibacterial Activity

The antifungal activity of *G. tridentata* against *Candida* strains is probably associated with the existence of the flavonoids, quercetin, and genistein derivates. Quercetin inhibits growth [65,66] and prevents microbial biofilm formation, possibly by reducing the cellular adhesion to abiotic surfaces, as described for *C. albicans* [67]. Also, genistein derivates inhibit *C. albicans* colony formation [68,69].

The antibacterial potential of *G. tridentata* can result from the higher content of phenolic compounds, flavonols, and isoflavones, which improves the antimicrobial power of extracts.

Additionally, the presence of taxifolin, genistin, and biochanin was previously reported as having an antibacterial activity [70,71,72], reducing bacteria resistance mechanisms and conducting delayed protein synthesis in *S. aureus* involved in the production of enzymes and nucleic acids required for bacterial growth [17,71,73]. As a result, membrane permeability increases, leading to a decline in bacteria survival. Also, flavonoids display a strong capacity to form complexes with bacteria cell walls, inhibiting bacteria growth [73,74,75].

Identically, rutin, isoquercitrin, and quercetin can act as antimicrobial agents, inhibiting nucleic acid and cytoplasmic membrane synthesis as well as bacterial metabolism [73]. Importantly, the above-mentioned phenolic compounds are more effective against Gram-positive bacteria [76], like *S. aureus*. Consequently, these compounds could have a bacteriostatic effect instead of a bactericidal action [17].

Regardless of the need for additional studies to clarify the mechanisms of action, it is possible to conclude that *G. tridentata* can be valuable in treating or complementing commercial drugs due to its bacteriostatic and antifungal potential. However, the extraction solvent influences the plant extracts bioactivity [37,39]. Thus, based on that, it could be important to evaluate the antifungal and antibacterial effects of other extraction methods of *G. tridentata*.

#### 4.2.4. Cytotoxicity

Firstly, it is important to clarify that human cell lines are preferred to primary cultures from animals, as they decrease species-related variations that can occur during the extrapolation of the results [77]. Secondly, data obtained by Serralheiro et al. (2013) [20], Ferreira et al. (2012) [6], and Caleja et al. (2019) [34] demonstrated that *G. tridentata* aqueous extracts could be appropriately used in traditional medicine, in cooking and the preservation of aliments against oxidative stress.

#### 4.2.5. Nematocidal/Nematotoxicity Activity

Although several studies showed that EOs display high nematocidal activities [78,79], literature about *G. tridentata* EOs is still limited. The mechanism of action of EOs is complex and occurs through different pathways, and some authors suggest that the interference of EOs in the nematode nervous system [80] is related to the neuromodulator octopamine [81] or GABA-gated chloride channels in insects [82]. Moreover, as in bacteria or fungi, EOs might disturb the cell membrane of the nematode by altering its permeability [80].

Importantly, the phytochemical profile of *G. tridentata* EOs (Table 2) demonstrated the presence of carvacrol and geraniol. Some studies demonstrated the nematocidal activity of carvacrol, facilitated via tyramine receptor, that triggered a signaling cascade and, by interacting with a receptor-like SER-2, led to nematode mortality [83]. Similarly, the activity of geraniol in membrane disturbing, changing membrane-bound protein, and the intracellular signaling pathways [84,85] has also been described. Although experimentally, EOs of *G. tridentata* display nematocidal activity; it is important to complement the in vitro tests with in vivo soil-based trials to validate the efficiency of this plant EOs.

The main advantages of the nematocidal/nematotoxicity activity of EOs are their minimal toxicity to mammalians and their low environmental persistence, making them safe and compatible as biological control agents [86].

#### 4.2.6. Other Activities

Results obtained by Paulo et al. (2008) [27] with the isolated compounds isoquercitrin and sissotrin revealed a time-dependent antihyperglycemic activity, indicating the presence of bioactive compounds that differently influence glucose uptake. While these results discourage the use of the plant in the control of glycemic blood levels, they highlight a possible post-prandial hypoglycemic effect of isoquercitrin. However, this work [27] presents several imperfections, especially regarding scientific work with animals, not revealing the number of animals used or blinding techniques.

Nevertheless, performing more studies to assess the antihyperglycemic activity of *G. tridentata* extracts is important to understand the mechanism of action of its bioactive compounds.

Serralheiro et al. (2013) [20] demonstrated using a colorimetric assay that the *G. tridentata* aqueous extracts display anti-acetylcholinesterase activity. Acetylcholine (ACh) is an important excitatory neurotransmitter responsible for peristaltic movements [87,88] and is also responsible for the regulation and establishment of an adequate environment for enzymatic digestion and absorption and the lubrication of intestinal material [89].

Serralheiro et al. (2013) [20] suggested that the flavonoids present in the extract may play an essential role in the inhibition of ACh by interacting with their active bind sites. Falé et al. (2012) [90] also showed small alterations in the ACh, namely in the aromatic amino acids, that affect enzymatic activity. The alterations occur in the presence of quercetin, romarinic acid, luteolin, and apigenin [90], all of which are also present in *G. tridentata* aqueous extracts.

Falé et al. (2014) [22] intended to understand the mechanism of action of *G. tridentata* extract in cholesterol in an in vivo experiment and found a reduction of 22% of plasma total cholesterol. Apart from the limited number of volunteers, the authors also do not refer to the control of other important variables such as diet and physical exercise.

They also performed in vitro studies to evaluate the Inhibitory concentration of HMG-CoA of *G. tridentata* extract HMG-CoA reductase. The downregulation of HMGCR is directly associated with cholesterol reduction via SREBP-2 activation as it acts in the upregulation of low-density lipoprotein receptor (LDLR) that increases the exclusion of cholesterol-rich low-density lipoprotein (LDL) particles from the blood circulation [91,92].

The same authors [22] also performed permeation studies in Caco-2 monolayer cells. The results demonstrated the bioavailability of some bioactive compounds, suggesting the biochanin A glucoside as a specific transporter of glycoside moiety that may be involved in the reduction of cholesterol [22]. Sadri et al. (2017) [93] also demonstrated that oral administration of biochanin A (10 and 15 mg/kg) in rats significantly decreased serum triglycerides, total cholesterol, and LDL cholesterol.

## 5. Conclusions

*Genista tridentata* extracts present a high extraction yield and have been described to possess high levels of flavonoids and phenolic compounds, suggesting a high potential for its application as a new source of natural antioxidants and preservatives for the food industry or in products with health benefits, such as nutraceuticals. Moreover, the results indicate that the plants can be collected at all seasons of the year, which represents an added benefit for the industry.

## Figures and Tables

**Figure 1 biology-12-01387-f001:**
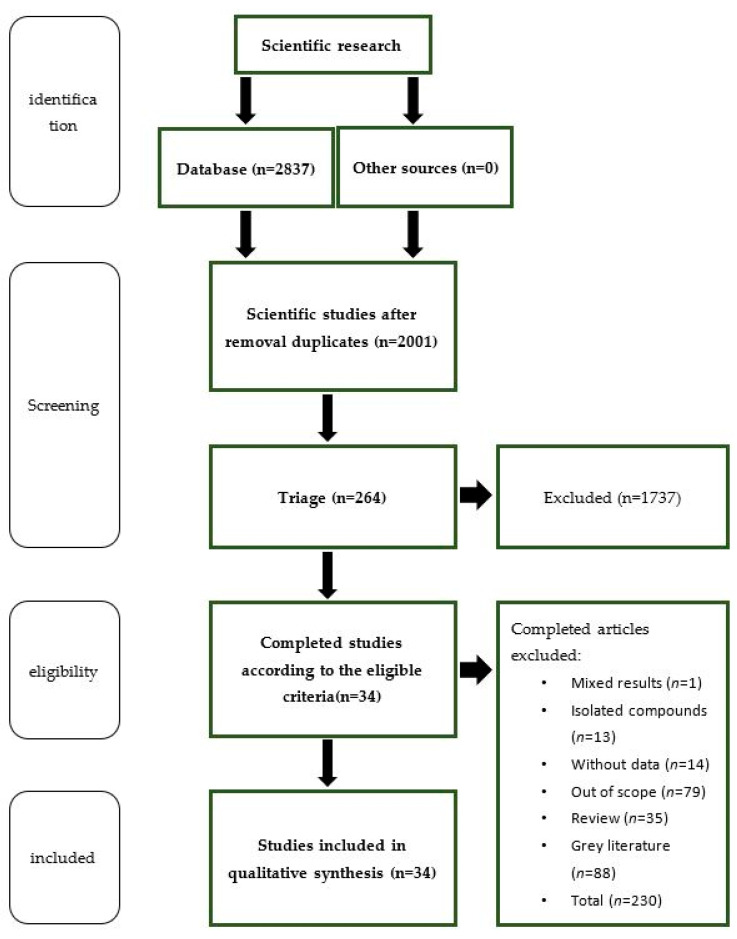
Schematic representation of the PRISMA flow chart depicting the literature screening process.

**Figure 2 biology-12-01387-f002:**
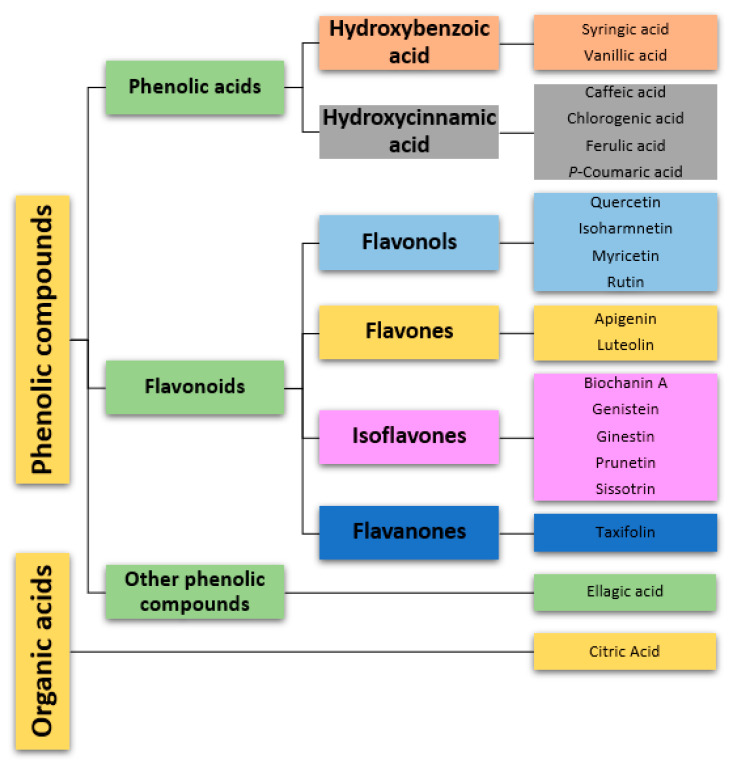
Classification of phytochemical compounds identified in *Genista tridentata*. Each distinct category is associated with a unique color.

**Table 1 biology-12-01387-t001:** Major biologically active compounds were found in several samples of *G. tridentata* (X—detected; NA—not available; ND—not detected). The table’s color scheme was aligned with Figure 2.

Authors	[17]	[23]		[9]	[15]	[20]	[24]		[27]	[32]	[35]	[24]						[6]	[22]	[15]	[35]	[43]
Extraction method	Methanolic							Aqueous												Ethanolic		NA
Part plant used	Crude	flowers	Stems and leaves	Flowers								Aerial parts				In vitro culture		Leaves + Flowers	NA	Flowers		Flowers
Sampling localization	Vila Real	Serra da Estrela	Serra da Estrela	Cinfães	Montesinho	Herbal Shop—DIÉTICA ^®^	Malcata	Gardunha	Cinfães	Herbal Shop—Ervital	Viseu	Malcata		Gardunha		Malcata	Gardunha	Herbal shop—Ervital	Herbal Shop—DIÉTICA ^®^	Montesinho	Viseu	Herbal shop—Ervital
Sampling period	NA				Spring	NA	May	May	NA	Spring 2012	NA	February	May	February	May	NA				Spring 2019	NA	Spring 2012
					2019																	
5,5′-Dihydroxy-3′-methoxy-isoflavone-7-O-β-glucoside	ND	ND	ND	X	ND	ND	X	X	ND	X	ND	X	X	X	X	X	X	ND	ND	ND	ND	X
5,5′-Dihydroxi-3′-methoxyisoflavone	ND	ND	ND	ND	X	ND	ND	ND	X	ND	ND	ND	ND	ND	ND	ND	ND	ND	ND	X	ND	ND
7-Methylorobol	ND	ND	ND	ND	ND	ND	X	ND	X	X	ND	ND	ND	X	X	X	ND	ND	ND	ND	ND	X
Apigenin 5,7-dimethyl	ND	ND	ND	ND	ND	X	ND	ND	ND	ND	ND	ND	ND	ND	ND	ND	ND	ND	ND	ND	ND	ND
Apigenin 5,7-dimethyl ether 4′galactoside	ND	ND	ND	ND	ND	ND	ND	ND	ND	ND	ND	ND	ND	ND	ND	ND	ND	ND	X	ND	ND	ND
Biochanin A	X	ND	ND	ND	ND	X	ND	ND	ND	X	ND	ND	ND	ND	ND	ND	ND	ND	X	ND	ND	X
Biochanin A O-acetylhexoside-O-hexoside	ND	ND	ND	ND	ND	ND	ND	ND	ND	X	ND	ND	ND	ND	ND	ND	ND	ND	ND	ND	ND	X
Biochanin A O-hexoside	ND	ND	ND	ND	ND	ND	ND	ND	ND	X	ND	ND	ND	ND	ND	ND	ND	ND	ND	ND	ND	X
Biochanin A O-hexoside-O-hexoside	ND	ND	ND	ND	ND	ND	ND	ND	ND	X	ND	ND	ND	ND	ND	ND	ND	ND	ND	ND	ND	X
Biochanin A-glucoside	X	ND	ND	ND	ND	X	ND	ND	ND	ND	ND	ND	ND	ND	ND	ND	ND	ND	X	ND	ND	ND
Caffeic acid	ND	X	X	ND	ND	ND	ND	ND	ND	ND	ND	ND	ND	ND	ND	ND	ND	ND	ND	ND	ND	ND
Chlorogenic acid	ND	ND	X	ND	ND	ND	ND	ND	ND	ND	ND	ND	ND	ND	ND	ND	ND	ND	ND	ND	ND	ND
Citric acid	ND	ND	ND	ND	ND	ND	X	X	ND	ND	ND	ND	X	X	X	X	X	ND	ND	ND	ND	ND
Dihydroquercetin 6-C-hesoxide	ND	ND	ND	ND	X	ND	ND	ND	ND	X	ND	ND	ND	ND	ND	ND	ND	ND	ND	X	ND	X
Ellagic acid	ND	X	X	ND	X	ND	ND	ND	ND	ND	ND	ND	ND	ND	ND	ND	ND	ND	ND	X	ND	ND
Ferulic acid	ND	X	X	ND	ND	ND	ND	ND	ND	ND	ND	ND	ND	ND	ND	ND	ND	ND	ND	ND	ND	ND
Genistein-8-C-glucoside	ND	ND	ND	ND	ND	X	X	X	ND	X	ND	X	X	X	X	X	X	ND	X	ND	ND	X
Ginestein	X	ND	ND	ND	ND	ND	X	X	X	X	X	X	ND	ND	X	X	ND	ND	ND	ND	X	X
Ginestein derivatives	X	ND	ND	ND	X	ND	ND	ND	ND	ND	X	ND	ND	ND	ND	ND	ND	ND	ND	X	X	ND
Ginestin	X	ND	ND	X	ND	ND	ND	ND	X	X	ND	ND	ND	ND	ND	ND	ND	ND	ND	ND	ND	X
Isoquercitrin	ND	ND	ND	X	ND	X	X	X	X	X	X	ND	X	ND	X	X	X	ND	X	ND	X	X
Isorhamnetin-O-hexoside	ND	ND	ND	ND	ND	ND	ND	ND	ND	ND	ND	ND	ND	ND	ND	ND	ND	X	ND	ND	ND	ND
Luteolin-O-(O-acetyl)-glucuronide	ND	ND	ND	ND	ND	ND	ND	ND	ND	ND	ND	ND	ND	ND	ND	ND	ND	X	ND	ND	ND	ND
Luteolin-O-glucuronide	ND	ND	ND	ND	ND	ND	ND	ND	ND	ND	ND	ND	ND	ND	ND	ND	ND	X	ND	ND	ND	ND
Methylbiochanin A/methylprunetin	ND	ND	ND	ND	ND	ND	ND	ND	ND	X	ND	ND	ND	ND	ND	ND	ND	ND	ND	ND	ND	X
Methylbiochanin A/methylprunetin derivative	ND	ND	ND	ND	ND	ND	ND	ND	ND	X	ND	ND	ND	ND	ND	ND	ND	ND	ND	ND	ND	X
Methylbiochanin A/methylprunetin O-hexoside	ND	ND	ND	ND	ND	ND	ND	ND	ND	X	ND	ND	ND	ND	ND	ND	ND	ND	ND	ND	ND	X
Myricetin-6-C-glucoside	ND	ND	ND	ND	X	X	X	X	X	X	ND	X	X	X	X	X	X	ND	X	X	-	X
p-Coumaric acid	ND	ND	X	ND	ND	ND	ND	ND	ND	ND	ND	ND	ND	ND	ND	ND	ND	ND	ND	ND	ND	ND
Pentahydroxy-flavonol-di-O-glucoside	ND	ND	ND	ND	ND	ND	ND	ND	ND	ND	ND	ND	ND	ND	ND	ND	ND	X	ND	ND	ND	ND
Prunetin	ND	ND	ND	X	ND	ND	X	ND	X	ND	ND	ND	ND	ND	X	ND	ND	ND	ND	ND	ND	X
Quercetin	ND	X	X	ND	ND	ND	ND	ND	ND	ND	ND	ND	ND	ND	ND	ND	ND	ND	ND	ND	ND	ND
Quercetin 3-O-galactoside	ND	ND	ND	ND	ND	ND	ND	ND	ND	ND	ND	X	ND	ND	X	ND	ND	ND	ND	ND	ND	ND
Quercetin deoxyhexosyl-hexoside	ND	ND	ND	ND	X	ND	ND	ND	ND	X	ND	ND	ND	ND	ND	ND	ND	ND	ND	X	ND	X
Quercetin O-hexoside	ND	ND	ND	ND	X	ND	ND	ND	ND	X	ND	ND	ND	ND	ND	ND	ND	ND	ND	X	ND	X
Quercetin-3-O-rutinoside	ND	ND	ND	ND	ND	ND	ND	ND	ND	X	X	ND	ND	ND	ND	ND	ND	ND	ND	ND	X	X
Quercetin derivates	ND	ND	ND	ND	X	ND	ND	ND	ND	ND	X	ND	ND	ND	ND	ND	ND	ND	ND	X	X	ND
Quinic acid	ND	ND	ND	ND	ND	ND	X	X	ND	ND	ND	X	X	X	X	X	X	ND	ND	ND	ND	ND
Rosmarinic acid	ND	ND	ND	ND	ND	ND	ND	ND	ND	ND	ND	ND	ND	ND	ND	ND	ND	X	ND	ND	ND	ND
Rutin	ND	ND	ND	ND	ND	ND	X	X	X	ND	ND	X	X	X	X	X	X	ND	ND	ND	ND	ND
Sissotrin	ND	ND	ND	X	ND	ND	X	X	X	X	ND	ND	ND	X	X	ND	ND	ND	ND	ND	ND	X
Syringic acid	ND	X	X	ND	ND	ND	ND	ND	ND	ND	ND	ND	ND	ND	ND	ND	ND	ND	ND	ND	ND	ND
Taxifolin	X	ND	ND	ND	ND	ND	ND	ND	ND	ND	ND	ND	ND	ND	ND	ND	ND	ND	ND	ND	ND	ND
Taxifolin-6-C-glucoside	ND	ND	ND	ND	ND	X	X	X	ND	ND	ND	X	X	X	X	ND	ND	ND	X	ND	ND	ND
Vanillic acid	ND	X	X	ND	ND	ND	ND	ND	ND	ND	ND	ND	ND	ND	ND	ND	ND	ND	ND	ND	ND	ND

**Table 2 biology-12-01387-t002:** Composition (%) of the essential oils of *G. tridentata* isolated by hydrodistillation, collected in different years and locations. (AMF02: Flowers, collected in Arneiro das Milhariças in 2002; AMF03: collected in Arneiro das Milhariças in 2003; AML02: collected in Arneiro das Milhariças in 2002; AML03: collected in Arneiro das Milhariças in 2003; PAPN: collected in Pedra de Altar, Proença a nova; PSFPN: collected in Póvoa, Sobreira Formosa, Proença a nova; SCB: collected in Sarzeda, Castelo Branco; MCSB: collected in Milhasa do Corvo, Sarzeda, Castelo Branco; ND—not detected).

Components		Flowers			Leaves + Stems		Aerial Parts				
	Authors	[11]		[29]	[11]						
	RI	AMF02	AMF03	Herbal Shop	AML02	AML03	PAPN	PFSPNa	PFSPNb	SCB	MCSB
*trans*-2-Hexenal	866	1.6	0.5	0.1	ND	1.6	ND	ND	1.7	3.2	ND
*cis*-3-Hexen-1-ol	868	1.6	1.2	ND	ND	5.3	ND	ND	0.8	3	ND
*cis*-2-Hexen-1-ol	882	1.5	1.2	ND	ND	0.8	ND	ND	0.6	1.2	ND
*n*-Hexanol	882	0.5	1.6	ND	ND	1.1	ND	ND	1.1	0.7	ND
*n*-Heptanal	897	11.8	4.8	0.9	ND	0.5	0.8	ND	ND	0.3	ND
*n*-Nonane	900	ND	ND	ND	ND	0.2	ND	ND	2.3	0.2	ND
Benzaldehyde	927	0.5	0.8	0.3	ND	0.6	1	ND	0.6	0.1	ND
α-Pinene	930	ND	0.3	0.3	ND	0.8	ND	ND	0.5	0.1	ND
*n*-Heptanol	952	0.5	1.6	ND	ND	1.5	ND	ND	ND	ND	1.3
1-Octen-3-ol	961	10.7	21	9.2	11.5	22.6	1.7	29.7	15	25.8	36.8
2-Pentyl furan	972	2.4	1.3	0.8	2.5	0.5	ND	ND	0.7	2.1	1.4
*n*-octanal	973	ND	ND	0.6	ND	ND	ND	ND	ND	ND	ND
3-Octanol	974	1.4	1.5	ND	1.9	ND	ND	ND	1.9	0.3	1.5
Benzyl alcohol	996	ND	ND	ND	0.3	0.4	ND	ND	ND	0.3	ND
Benzene acetaldehyde	1002	1.8	1.8	ND	0.3	1.2	ND	ND	0.4	1.4	0.6
*ρ*-Cymene	1003	ND	ND	0.3	ND	ND	ND	ND	ND	ND	ND
1,8-Cineole	1005	0.9	1	0.7	1.1	0.2	ND	ND	ND	ND	ND
Limonene	1009	0.9	1	ND	1.1	0.2	ND	ND	0.3	ND	ND
Acetophenone	1017	ND	1.4	ND	2.1	0.5	ND	ND	ND	ND	ND
*n*-Octanol	1045	0.5	0.4	0.7	2.1	0.3	0.6	ND	ND	ND	ND
*ρ*-Cymenene	1050	ND	ND	0.6	ND	ND	ND	ND	ND	ND	ND
Heptanoic acid	1056	0.5	1.2	ND	ND	0.4	ND	ND	ND	ND	2.1
Phenyl ethyl alcohol	1064	0.7	1.2	ND	2	1.7	ND	3.6	3.3	3.4	6.3
*n*-Nonanal	1073	14.5	6.1	6.5	4.6	0.9	10.5	4.1	0.2	0.9	1
Linalol	1074	2.9	0.5	7.1	ND	2	ND	5.2	ND	2.3	1
*cis*-Rose oxide	1083	2.9	0.5	ND	ND	ND	2	ND	5.2	2.3	1
Camphor	1095	ND	ND	0.7	ND	ND	ND	ND	ND	ND	ND
*n*-Undecane	1100	ND	ND	ND	ND	ND	ND	1	2.3	0.2	ND
*trans*-Rose oxide	1100	ND	ND	ND	2.1	0.7	ND	1	ND	ND	ND
*trans*-Pinocarveol	1106	ND	ND	0.3	ND	ND	ND	ND	ND	0.2	ND
2- *trans*,6 *cis*-Nonadienal	1106	2.1	0.3	0.2	ND	ND	ND	ND	ND	ND	ND
2- *trans*-Nonen-1-al	1114	0.5	0.4	ND	2.2	0.2	ND	ND	ND	ND	ND
Pentyl benzene	1119	1.5	ND	ND	ND	0.3	ND	ND	ND	ND	ND
Menthone	1120	ND	ND	0.2	ND	ND	ND	ND	ND	ND	ND
Benzyl acetate	1123	ND	ND	0.2	ND	ND	ND	ND	ND	ND	ND
Borneol	1134	ND	ND	1.1	ND	ND	ND	ND	ND	ND	ND
Lavandulol	1142	ND	ND	0.3	ND	ND	ND	ND	ND	ND	ND
Menthol	1148	ND	ND	0.5	ND	ND	ND	ND	ND	ND	ND
Terpinen-4-ol	1148	ND	ND	0.7	ND	ND	ND	ND	ND	ND	ND
Octanoic acid	1156	0.3	ND	0.5	0.5	ND	ND	ND	ND	ND	ND
α-Terpineol	1159	ND	ND	1.8	ND	ND	ND	1.2	0.8	0.3	ND
Safranal	1160	1.4	0.3	ND	ND	0.5	ND	ND	ND	ND	ND
Methyl chavicol (=estragole)	1163	ND	ND	0.9	ND	ND	ND	ND	ND	ND	ND
*n*-Decanal	1180	ND	0.3	0.4	ND	ND	ND	ND	ND	ND	ND
Pulegone	1210	ND	ND	1.4	ND	ND	ND	ND	ND	ND	ND
Geraniol	1236	0.3	1.6	0.6	4	9.2	3.2	1	-	1.4	2.8
Linalyl acetate	1245	ND	ND	1.4	ND	ND	ND	ND	ND	ND	ND
*Trans*-Anethole	1254	ND	ND	4.7	ND	ND	ND	ND	ND	ND	ND
*n*-Decanol	1259	0.3	1.6	0.6	4	0.2	3.2	3.4	2.5	3.2	1.9
2-Undecanone	1273	ND	ND	2.2	ND	ND	ND	ND	ND	ND	ND
Perilla alcohol	1274	ND	ND	ND	ND	3.4	ND	ND	ND	0.6	ND
Nonanoic acid	1274	ND	0.3	1.5	2.3	ND	ND	ND	ND	ND	ND
*cis*-Theaspirane	1279	1.6	2.2	ND	12.7	7.1	14.2	5.3	13.2	9	6.2
2 *trans*,4 *trans*-Decadienal	1285	0.8	1.3	ND	ND	0.1	ND	1.8	ND	2	ND
*cis*-Transpirane	1286	ND	ND	3.2	ND	ND	ND	ND	ND	ND	ND
Carvacrol	1286	ND	ND	0.3	ND	ND	ND	ND	ND	ND	ND
2-*trans*-4-*trans*-Decadienal	1286	ND	ND	1	ND	ND	ND	ND	ND	ND	ND
*trans*-Theaspirane	1300	2.4	1.9	3.9	12.1	6.8	17.2	6.3	13.6	10	5.5
Hexyl tiglate ester	1316	ND	ND	0.2	ND	ND	ND	ND	ND	ND	ND
Eugenol	1327	1.4	1.7	0.8	3.5	2.6	ND	3.1	3	3.2	3.6
α-Terpenyl acetate	1334	ND	ND	0.3	ND	ND	ND	ND	ND	ND	ND
α-Longipinene	1338	ND	ND	0.1	ND	ND	ND	ND	ND	ND	ND
Decanoic acid	1350	ND	ND	0.8	ND	ND	ND	ND	ND	ND	ND
*trans*-β-Dasmascenone	1356	ND	ND	0.8	ND	ND	ND	ND	ND	ND	ND
Geranyl acetate	1370	ND	ND	0.5	ND	ND	ND	ND	ND	ND	ND
α-Copaene	1375	ND	ND	ND	ND	ND	ND	0.9	ND	ND	ND
β-Bourbonene	1379	ND	ND	ND	ND	ND	ND	1.5	ND	1.1	ND
2-Pentadecanone	1390	ND	ND	0.8	ND	ND	ND	ND	ND	ND	ND
Longifolene	1399	ND	ND	ND	ND	ND	ND	1.4	ND	ND	ND
β-Caryophyllene	1414	ND	0.4	1.2	ND	ND	ND	2.7	ND	2	0.9
Geranyl acetonea	1434	ND	3.6	0.7	ND	ND	ND	1.2	ND	0.6	ND
allo-Aromadendrene	1456	ND	ND	0.7	ND	ND	ND	ND	ND	ND	ND
*trans-β*-Ionone	1456	ND	ND	1.1	ND	ND	ND	ND	ND	ND	ND
Germacrene-D	1474	ND	0.2	ND	ND	ND	9.7	3.3	ND	0.7	ND
α-Curcumene	1475	ND	ND	0.5	ND	ND	ND	ND	ND	ND	ND
ƴ-Cadinene	1500	ND	3.3	ND	ND	ND	ND	1.2	ND	1.1	1.9
σ-Cadinene	1505	ND	2.4	ND	ND	ND	ND	1.6	ND	2	1.9
Dodecanoic acid	1551	3.5	2.1	5.3	2.6	0.3	15	ND	ND	0.9	1.1
β-Caryophyllene oxide	1561	ND	ND	ND	ND	ND	ND	1.3	ND	1.2	2.9
*n*-Tetradecanal	1596	ND	ND	ND	ND	ND	ND	1.1	ND	2.7	1.5
*n*-Pentadecanal;	1688	ND	ND	ND	ND	ND	ND	ND	ND	0.8	ND
Tetradecanoic acid	1734	ND	ND	0.2	ND	ND	ND	ND	ND	ND	ND
Hexadecanoic acid	1779	ND	ND	0.7	ND	ND	ND	ND	ND	ND	ND
9,12-Octadecadienoic acid	1820	ND	ND	0.4	ND	ND	ND	ND	ND	ND	ND
% of identified components		71.8	75.1	71.8	78.4	76.8	77.1	82.9	64.8	88.5	82.2
Grouped components											
Monoterpene hydrocarbons		0.9	1.3	0.6	1.1	1	ND	ND	0.8	0.1	ND
Oxygen-containing monoterpenes		6.2	7	18.6	10.6	17.5	3.2	9.6	0.8	5.4	3.8
Sesquiterpene hydrocarbons		ND	6.3	2.5	ND	ND	9.7	12.6	ND	6.9	4.7
Oxygen-containing sesquiterpenes		ND	ND	7.1	ND	ND	ND	1.3	ND	1.2	2.9
Phenylpropanoids		1.4	1.7	6.4	3.5	2.6	ND	3.1	3	3.2	3.6
Oil yield (*v*/*w*)		<0.05%	<0.05%	0.01%	<0.05%	<0.05%	<0.05%	<0.05%	<0.05%	<0.05%	<0.05%

**Table 3 biology-12-01387-t003:** Antioxidant activity and phenolic and flavonoid content in several samples of *G. tridentata*. (AA—ascorbic acid; AAE—ascorbic acid equivalent; ext—extract; GAE- gallic acid equivalent; ip—inhibition percentage; CAE—caffeic acid equivalent; ClAE—chlorogenic acid equivalent; inf—infusion; Pt—Portugal; Sp—Spain; T—Trolox. TAC—total antioxidant capacity; in cases where the units are not indicated, the unit indicated in the first line must be considered; NA—not available; NR—not reported).

Authors	Extraction	Part Plant Used	Localization	Sampling Period	DPPH IC50	Lipidic Peroxidation (TBARS)	Total Phenol Content (mg GAEg^−1^ dw)	Total Flavonoids Content (QE mg/g dw)
[17]	Methanolic	Crude	Vila Real. Pt	NA	NA	NA	3.664 ± 0.04 mg g^−1^ dw	NA
[12]		Flowers	Trás-os-Montes. Pt	Spring 2010 (Freeze-drying)	NA	0.12 ± 0.02 mg/mL	523.42 ± 36.09 mg ClAE/g ext	58.12 ± 5.78
			Trás-os-Montes. Pt	Spring 2010 (Shade-drying)	NA	0.13 ± 0.04 mg/mL	519.81 ± 40.24 mg ClAE/g ext	85.58 ± 5.60
[23]			Serra da Estrela. Pt	NA	26.1 ± 1.3 mg/L	NR	171.4 ± 0.7	NR
[36]			Herbal Shop—Ervital	Spring 2012	NR	1.18 ± 0.06 mg/mL	NR	NR
[23]		Stems and leaves	Serra da Estrela. Pt	NA	69.7 ± 11.9 mg/L	NR	113.6 ± 1.5	NR
[13]	Aqueous	Crude	Herbal Shop—Ervital	NA	NR	NR	0.308 ± 0.004 (g L^−1^ GAE)	NR
[5]		Flowers	Orvalho Mountain. Pt	May	3.6 ± 0.03 mMT/Kg dw	NR	402.9 ± 17.07	NR
			Gardunha Mountain. Pt		3.2 ± 0.14 mMT/Kg dw	NR	337.7 ± 50.83	NR
			Malcata Mountain. Pt		3.5 ± 0.03 mMT/Kg dw	NR	309.5 ± 19.82	NR
[15]			Montesinho. Pt	Spring 2019	NR	IC50 (μg/mL): 5.3 ± 0.1	NR	NR
[20]			Herbal Shop—Dietética	NA	IC50 (μg/mL): 18.6 ± 0.7	NR	NR	NR
[34]			Herbal Shop—Ervital	NA	NR	8.4 ± 0.2 μg/mL	107 ± 2 (mg/g)	107 ± 2 (mg/g)
[35]			Viseu. Pt	NA	IC50 (μg/mL): 158 ± 1.45	IC50 (μg/mL): 83.48 ± 6.17	34.80 mg/g of dried extract	NR
[16]		Aerial Parts	Algarve. Pt	Spring 2012 (Cold)	NR	NR	314.89 ± 47.49 (μmol GAE gdw^−1^)	NR
				Spring 2012 (Hot)	NR	NR	529.35 ± 3.01 (μmol GAE gdw^−1^)	NR
[5]		Stems (dormancy period)	Orvalho Mountain. Pt	January	3.6 ± 0.07 mMT/Kg dw	NR	331.7 ± 35.05	NR
			Gardunha Mountain. Pt		3.2 ± 0.07 mMT/Kg dw	NR	394 ± 74.5	NR
			Malcata Mountain. Pt		3.5 ± 0.08 mMT/Kg dw	NR	320 ± 70.23	NR
		Stems (flowering period)	Orvalho mountain. Pt	May	3.6 ± 0.01 mMT/Kg dw	NR	335.9 ± 34.59	NR
			Gardunha Mountain. Pt		3.2 ± 0.07 mMT/Kg dw	NR	270.7 ± 70.8	NR
			Malcata Mountain. Pt		3.6 ± 0.05 mMT/Kg dw	NR	315.8 ± 73.5	NR
[13]		Leaves	Herbal Shop—Ervital	NA	NR	NR	0.130 ± 0.026 (g L^−1^ GAE)	NR
[23]		NA	Serra da Estrela. Pt	NA	NR	NR	222.69 ± 5.12	NR
[28]			Serra da Estrela. Pt	NA	42.97 ± 1.69 (IC50 mg/L)	NR	-	NR
[30]			Herbal Shop—Ervital	NA	NR	NR	44.1 ± 0.6 (CAE. mg g^−1^ dw)	NR
[32]			Herbal Shop—Ervital	Spring 2012	NR	NR	NR	33.40 ± 0.28 mg/g
[15]	Ethanolic	Flowers	Montesinho. Pt	Spring 2019	NR	IC50 (μg/mL): 3.19 ± 0.02	NR	NR
[35]			Viseu. Pt	NA	IC50 (μg/mL): 115 ± 0.70	IC50 (μg/mL): 113 ± 15.53	42.84 mg/g of dried extract	NR
[6]		Herbal Shop—Ervital	NA	NA	NR	NR	15.5 ± 16.5	NR
[28]		NA	Serra da Estrela. Pt	NA	60.39 ± 1.78 (IC50 mg/L)	NR	196.61 ± 3.94	NR

**Table 4 biology-12-01387-t004:** Antifungal and antibacterial effect of *G. tridentata* extract. (IZ: inhibitory zone; When there was no evident halo, but inhibition of growth, the effect was also classified as (+) cell growth inhibition, (++) cell density reduction, (+++) cell density reduction and growth inhibition; MIC: minimal inhibitory concentration if MIC values ≤ 100 µg.mL^−1^, moderate (++) when 100 < MIC ≤ 500 µg.mL^−1^, not detectable (-) <4 g/L^−1^; NR—not reported).

Authors	Extraction	Part Plant Used	Species	Strains	Method	Results
[37]	Methanolic	Flowers	*Candida albicans*	ATCC90028	Disc diffusion assay	IZ: 10 mm
				575541		-
				557834		IZ: 10 mm
				558234		IZ: 9 mm
			*Candida galabrata*	ATCC2001		IZ: 11 mm
				D1		IZ: 11 mm
				513100		IZ: 9 mm
			*Candida parapsilosis*	ATCC22019		++
				AM2		++
				AD		-
				491861		-
				513143		-
			*Candida tropicalis*	ATCC750		+
				AG1		+++
				75		-
				12		-
				544123		-
				519468		-
				T2.2		-
[34]	Aqueous	Flowers	*Aspergillus niger*	ATCC 6275	Microdilution method	MIC: 8 mg/mL
			*Aspergillus versicolor*	ATCC 11730		MIC: 0.5 mg/mL
			*Penicillium funiculosum*	ATCC 36839		MIC: 0.5 mg/mL
			*Penicillium verrucosum*	Food isolates		MIC: 0.5 mg/mL
[15]			*Aspergillus niger*	ATCC 6275		MIC: 0.5 mg/mL
			*Aspergillus versicolor*	ATCC 11730		MIC: 0.5 mg/mL
			*Aspergillus fumigatus*	Human isolate		MIC: 1 mg/mL
			*Penicillium funiculosum*	ATCC 26839		MIC: 0.5 mg/mL
			*Penicillium aurantiogriseum*	ATCC 58604		MIC: 0.5 mg/mL
	Ethanolic	Flowers	*Aspergillus niger*	ATCC 6275		MIC: 0.25 mg/mL
			*Aspergillus versicolor*	ATCC 11730		MIC: 0.25 mg/mL
			*Aspergillus fumigatus*	Human isolate		MIC: 0.25 mg/mL
			*Penicillium funiculosum*	ATCC 26839		MIC: 0.25 mg/mL
			*Penicillium aurantiogriseum*	ATCC 58604		MIC: 0.5 mg/mL
[17]	Methanolic	Crude	*Staphylococcus aureus*	ATCC 13565	Microdilution method	MIC: 312.5 µg.mL^−1^ (moderate)
				MJMC021		MIC: 78.1 µg.mL^−1^ (strong)
				MJMC024		MIC: 78.1 µg.mL^−1^ (strong)
				MJMC026		MIC: 78.1 µg.mL^−1^ (strong)
				MJMC025		MIC: 39.1 µg.mL^−1^ (strong)
				MJMC027		MIC: 39.1 µg.mL^−1^ (strong)
				MJMC029		MIC: 39.1 µg.mL^−1^ (strong)
[38]				*S. aureus* CECT 97	Disc diffusion test	MIC < 4 g/L^−1^ (indifferent)
[39]		Flowers		ATCC 25923	Disc diffusion test	Inhibitory zone: 5 mm
[15]	Aqueous	Flowers		ATCC 6538	Microdilution method	MIC: 0.25 mg/mL
[34]			*Escherichia coli*	NR	Microdilution method	MIC: 0.5 mg/mL
			*Salmonela ryphimurium*	NR		MIC: 1 mg/mL
			*Bacillus cereus*	NR		MIC: 1 mg/mL
			*Listeria monocytogenes*	NR		MIC: 1 mg/mL
[15]			*Micrococcus flavus*	ATCC 10240		MIC: 2 mg/mL
			*Enterobacter cloacae*	ATCC 35030		MIC: 1 mg/mL
			*Bacillus cereus*	Clinical isolate		MIC: 1 mg/mL
			*Listeria monocytogenes*	NCTC 7973		MIC: 1 mg/mL
			*Salmonella typhimurium*	ATCC 13311		MIC: 1 mg/mL
	Ethanolic		*Staphylococcus aureus*	ATCC 6538		MIC: 0.25 mg/mL
			*Micrococcus flavus*	ATCC 10240		MIC: 1 mg/mL
			*Enterobacter cloacae*	ATCC 35030		MIC: 1 mg/mL
			*Bacillus cereus*	Clinical isolate		MIC: 0.5 mg/mL
			*Listeria monocytogenes*	NCTC 7973		MIC: 0.5 mg/mL
			*Salmonella typhimurium*	ATCC 13311		MIC: 0.5 mg/mL

**Table 5 biology-12-01387-t005:** Evaluation of the quality of included articles using a modified version ^§^ of the ARRIVE guidelines for the reporting of in vivo experiments. (Criteria 1—Title; 2—Abstract; 3—Background; 4—Objectives; 5—Ethical statement; 6—Study design; 7—Inclusion/Exclusion criteria; 8—Experimental procedures; 9—Randomization; 10—Blinding; 11—Plant sample characterization ^§^; 12—Plant extraction methods ^§^; 13—Experimental outcomes; 14—Experimental animals; 15—Housing and husbandry; 16—Sample size; 17—Statistical methods; 18—Results; 19 –Interpretation/scientific implications; 20—Funding; Score between 0–1 (0—Absent (orange); 0.5—Incomplete or not applicable (yellow); 1—Complete (green)).

References	1	2	3	4	5	6	7	8	9	10	11	12	13	14	15	16	17	18	19	20	Score	Rating
[4]																					17	Strong
[5]																					16	Strong
[6]																					16	Strong
[9]																					16	Strong
[11]																					16	Strong
[12]																					16	Strong
[13]																					16	Strong
[14]																					13	Moderate
[15]																					16	Strong
[16]																					16	Strong
[17]																					16	Strong
[18]																					15	Moderate
[19]																					16	Strong
[20]																					16	Strong
[22]																					14	Moderate
[23]																					15	Moderate
[24]																					14	Moderate
[27]																					13	Moderate
[28]																					13	Moderate
[29]																					15	Moderate
[30]																					16	Strong
[31]																					14	Moderate
[32]																					15	Moderate
[33]																					14	Moderate
[34]																					16	Strong
[35]																					17	Strong
[36]																					15	Moderate
[37]																					15	Moderate
[38]																					16	Strong
[39]																					16	Strong
[40]																					15	Moderate
[41]																					13	Moderate
[42]																					14	Moderate
[43]																					15	Moderate

## Data Availability

The data presented in this study are available on request from the corresponding author.

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
