# Peer review of "Genista tridentata Phytochemical Characterization and Biological Activities: A Systematic Review"

_biology, 2023, doi:10.3390/biology12111387_

Round 1
Reviewer 1 Report
Comments and Suggestions for Authors
The manuscript consists of a review of the phytochemical characterization and biological activities of Pterospartum tridentatum. In my opinion, the manuscript is interesting and is appropriate for publishing in the journal. However, discussing the results with those obtained in at least one previous non-systematic review (https://doi.org/10.3390/medicines7060031) is necessary to conclude more accurately.
Author Response
- Summary
We would like to thank very much for the reviewer’s effort and dedication in evaluating our manuscript. Please find the detailed responses below and the corresponding revisions/corrections highlighted in the re-submitted files. The points addressed by the reviewer allowed us to considerably improve our work.
- General Evaluation
|
Questions for General Evaluation |
Reviewer’s Evaluation |
Response and Revisions |
|
Is the work a significant contribution to the field? |
We appreciate the grade, and we are confident that our systematic review holds significant value for fellow researchers and the broader scientific community. |
|
|
Is the work well organized and comprehensively described? |
Thank you |
|
|
Is the work scientifically sound and not misleading? |
Thank you |
|
|
Are there appropriate and adequate references to related and previous work? |
Thank you for the grade. We have made improvements and incorporated new references. Parte superior do formulárioParte inferior do formulário |
|
|
Is the English used correct and readable? |
Thank you |
- Point-by-point response to Comments and Suggestions for Authors
Comment #1: The manuscript consists of a review of the phytochemical characterization and biological activities of Pterospartum tridentatum. In my opinion, the manuscript is interesting and is appropriate for publishing in the journal. However, discussing the results with those obtained in at least one previous non-systematic review (https://doi.org/10.3390/medicines7060031) is necessary to conclude more accurately.
Response: In accordance with the reviewer’s suggestion, we have now included the non-systematic review (new reference [3]) in the discussion and introduction sections. We also included references [4, 7 and 8] to improve the “Introduction” and “Discussion” sections that now read:
Lines 56-65: "Genista tridentata (L.) Willk. or Pterospartum tridentatum (L.) Willk. (Genista tridentata (L.) is the recognized name for this species, and Pterospartum tridentatum (L.) Willk. is the commonly used name in both scientific literature and commercially available extracts. Among other synonyms, Chamaespartum tridentatum (P.) Gibbs is also used [3,4]), commonly known as “prickled broom”, is a Leguminosae (Fabaceae) species belonging to the subfamily Papilionoideae [5,6]. In line with scientific literature and the Global Biodiversity Information Facility database [7], the recorded countries of origin for the plant remain consistent, comprising Portugal, Spain, and Morocco. However, it's important to mention that the Botanic Royal Garden database [8] also lists Algeria and Tunisia as potential countries of origin for this plant.”
Lines 385: “The variability of assays facilitates the establishment of G. tridentata potential [3].”
Lines 422-424: “The phytochemical characterization of G. tridentata extracts demonstrated their richness in flavonoids, frequently correlated with anti-inflammatory activity [3,54].”
Lines 427-430: " Pinto et al. (2020) [3] provided a comprehensive review that explores the potential mechanisms of action of several bioactive compounds also found in G. tridentata concerning their anti-inflammatory activity.”
Reviewer 2 Report
Comments and Suggestions for Authors
Pterospartum tridentatum is a synsonym of Genista tridentata L. it will be better if you use the new generic and specific binomial nomenclature of the plant. Please revise this point both for Abstract section and Introduction
According to POWO (https://powo.science.kew.org/taxon/urn:lsid:ipni.org:names:496422-1#synonyms) the plant species is native to Algeria, Morocco, Portugal, Spain, Tunisia NOT endemic to Europe
It will be better if the authors include all phytochemicals (pure compounds) isolated from the plant species
There a pure compounds evaluated for bioactivities? There is a patent from the plant species
Author Response
|
Questions for General Evaluation |
Reviewer’s Evaluation |
Response and Revisions |
|
Is the work a significant contribution to the field? |
We appreciate the grade, and we are confident that our systematic review holds significant value for fellow researchers and the broader scientific community. |
|
|
Is the work well organized and comprehensively described? |
Thank you. |
|
|
Is the work scientifically sound and not misleading? |
Thank you. |
|
|
Are there appropriate and adequate references to related and previous work? |
Thank you.Parte superior do formulárioParte inferior do formulário |
|
|
Is the English used correct and readable? |
Thank you. |
- Point-by-point response to Comments and Suggestions for Authors
Comment #1: Pterospartum tridentatum is a synonym of Genista tridentata L. it will be better if you use the new generic and specific binomial nomenclature of the plant. Please revise this point both for Abstract section and Introduction
Response: Taxonomy can indeed be a complex and controversial field in biology, as different researchers may have varying opinions on the classification and naming of species. In the case mentioned, it seems there is some disagreement regarding the scientific name of the plant species. Because of that, we used the three different scientific names of the plant in our literature search: "Pterospartum tridentatum", "Chamaespartium tridentatum" and "Genista tridentata.". We opted to use the scientific nomenclature Pterospartum tridentatum (L.) Willk. Since this name is the most used by some taxonomists and researchers to refer to the plant in question. However, according to the Plant List database, both "Pterospartum tridentatum" and "Chamaespartium tridentatum" are considered synonyms of "Genista tridentata." Thus, based on the information available in Plant List database, "Genista tridentata L." appears to be the most widely accepted and currently recognized scientific name for this plant species. Therefore, in accordance with the reviewer’s suggestion, we altered the text of the manuscript which now reads “Genista tridentata”. We also included the following explanation in the introduction section: (Lines 56-61) that now read “Genista tridentata (L.) Willk. or Pterospartum tridentatum (L.) Willk. (Genista tridentata (L.) Willk. is the recognized name for this species, and Pterospartum tridentatum (L.) Willk. is the commonly used name in both scientific literature and commercially available extracts. Among other synonyms, Chamaespartum tridentatum (P.) Gibbs is also used [3,4]), commonly known as “prickled broom”, is a Leguminosae (Fabaceae) species belonging to the subfamily Papilionoideae [5,6].”
Comment #2: According to POWO (https://powo.science.kew.org/taxon/urn:lsid:ipni.org:names:496422-1#synonyms) the plant species is native to Algeria, Morocco, Portugal, Spain, Tunisia NOT endemic to Europe.
Response: None of the articles that were consulted and/or used in this systematic review make any reference to the plant being autochthonous to Algeria and Tunisia. Moreover, the information relating to the plant species' origin was corroborated through the Global Biodiversity Information Facility database. However, upon examination of the provided link, it was observed that Algeria and Tunisia are also listed as the countries of origin for the plant. Unable to challenge this assertion, we have included an additional paragraph in the introduction to recognize this inconsistency. Lines 61-65, that now read: “In line with scientific literature and the Global Biodiversity Information Facility database [7], the recorded countries of origin for the plant remain consistent, comprising Portugal, Spain, and Morocco. However, it's important to mention that the Plants of the World Online (POWO) database [8] also lists Algeria and Tunisia as potential countries of origin for this plant.”.
Comment #3: It will be better if the authors include all phytochemicals (pure compounds) isolated from the plant species.
Response: In accordance with exclusion criterion IV, which specifies "iv) reports with commercial pure compounds not directly derived from plant biomass," we only have included articles that discuss pure compounds isolated from plant species while excluding articles that refer to commercially produced pure compounds. Among the eligible articles for the systematic review, two of them discuss isolated bioactive compounds: Vitor et al. (2004) that isolated four isoflavones (5,5'-dihydroxy-3'-metoxi-isoflavone-7-O-beta-glucoside, prunetin, genistin and sissotrin) (page 7, line 233-235) and Paulo et al. (2008) that isolated and tested the effects of isoquercetin and sissotrin in the oral glucose tolerance in rats (page 8-9, line 320-324).
While we believe our work clearly stated the work of Vitor et al., (2004) was performed with isolated compounds, we felt the need to clarify that the article of Paulo et al. (2008) also used isolated compounds, so the discussion section has been modified accordingly and now reads: “Results obtained by Paulo et al. (2008) [28] with the isolated compounds isoquercitrin and sissotrin, revealed a time-dependent antihyperglycemic activity (…)” (Lines 492-494).
Comment #4: There a pure compounds evaluated for bioactivities? There is a patent from the plant species.
Response: As mentioned in our answer to the previous comment, we only included/analyzed data involving pure compounds when these were isolated from plant extracts and not synthetized artificially in a lab.
As far as we know and after performing several searches using both “Genista tridentata” and “Pterospartum tridentatum” as keywords, we found no patents associated with this plant. There were a few entries that hinted to this possibility however when the documents were analyzed we could not confirm the patent. We would like to ask the reviewer for help finding this information.
Reviewer 3 Report
Comments and Suggestions for Authors
The authors systematically review the research advance on the bioactivities and phytochemical profile of Pterospartum tridentatum by searching the literature to understand its use in folk medicine and its pharmacological potential. The manuscript might be published in the journal Biology after major revision.
1. From a total of 264 potentially eligible studies considered for screening, only 34 papers were considered eligible for this systematic review according the authors’ Inclusion and exclusion criteria. But the papers cited a total of 91 papers. How to explain it?
2. All the Latin names including plants, microorganism, such as those in lines 254, 259-263, etc. should be italic.
3. In “the highest values (0.260 ± 0.030gL-1” in line 198, and “O2-” in line 207-208,”-1” and “2-” should be superscript. Check the percent in “with gastric (IC50= 105.8±5.5 %) and pancreatic juices (IC50= 103.5±17.3 %)”. 50 should be subscript.
4. In table 1, the name of some compounds is not intact, such as 7-beta-glucoside in line 8. Some names didn’t showed the distinct structure, such as “hexoside”, “derivatives”. In table 2, check the name with “a-”, “b-”, “c-” or “d” which should be Greek letter alfa or beta etc. In table 5, please give the meaning of the color.
5. Check the format of references, some of them aren’t normative. The Latin names including plants and microorganism should be italic. Only the first letter of the first word in the title should be capital, while others are lowercase, such as ref. 33, 44, 52, 61, 70, 91, etc. All the first letter in the journals should be capital such as ref. 43, 53, 55, etc.
Author Response
- Summary
We would like to thank very much for the reviewer’s effort and dedication in evaluating our manuscript. Please find the detailed responses below and the corresponding revisions/corrections highlighted in the re-submitted files. The points addressed by the reviewer allowed us to considerably improve our work.
- General Evaluation
|
Questions for General Evaluation |
Reviewer’s Evaluation |
Response and Revisions |
|
Is the work a significant contribution to the field? |
Thanks for the grade, we are confident that our systematic review holds significant value for fellow researchers and the broader scientific community. |
|
|
Is the work well organized and comprehensively described? |
Thanks for the grade. We followed the PRISMA guidelines to systematic reviews. |
|
|
Is the work scientifically sound and not misleading? |
Thanks for the grade. We followed the PRISMA guidelines to systematic reviews. |
|
|
Are there appropriate and adequate references to related and previous work? |
Thanks for the grade. We have made improvements and incorporated new references.Parte superior do formulárioParte inferior do formulário |
|
|
Is the English used correct and readable? |
Thanks for the grade. We revised the manuscript taking in account your evaluation. |
- Point-by-point response to Comments and Suggestions for Authors
Comment #1: From a total of 264 potentially eligible studies considered for screening, only 34 papers were considered eligible for this systematic review according to the authors’ Inclusion and exclusion criteria. But the papers cited a total of 91 papers. How to explain it?
Response: The systematic review included a total of 34 eligible articles, which were analyzed, and their findings were presented in the “Results” section. Additionally, the quality of these articles was evaluated (table 5). However, an additional 57 articles were referenced in the “Introduction” and “Discussion” sections to provide an adequate background and to improve our understanding of the results obtained in the systematic review.
Comment #2: All the Latin names including plants, microorganism, such as those in lines 254, 259-263, etc. should be italic.
Response: We thank the referee for this important corrections and remarks, as requested, the text was altered to italics whenever referring to a scientific name.
Comment #3: In “the highest values (0.260 ± 0.030gL-1” in line 198, and “O2-” in line 207-208,”-1” and “2-” should be superscript. Check the percent in “with gastric (IC50= 105.8±5.5 %) and pancreatic juices (IC50= 103.5±17.3 %)”. 50 should be subscript.
Response: We appreciate to the reviewer for these valuable corrections and comments, the text has been modified accordingly. We also check the percentages mentioned in our systematic-review and the values are in accordance with the results obtained by Serralheiro et al., (2013) (present in table 4 of the article with reference [20]).
Comment #4: In table 1, the name of some compounds is not intact, such as 7-beta-glucoside in line 8. Some names didn’t show the distinct structure, such as “hexoside”, “derivatives”.
Response: We thank the referee for this important corrections and remarks. The mention of 7-beta-glucoside comes directly from the original paper. We agree that some missing part exist for this compound. We checked in the original paper, and other related papers with this species, but it was impossible to determine what is the missing part. Taking this in consideration, and that the original is already wrong, we eliminated this line from the Table 1 to avoid spread of misleading information.
Regarding the structures, we understand and agree that the term "hexoside" is vague and could indicate different 6C-glucosides with different structure, but this is how it is named in the original papers. In this case we think we should maintain the term, since the information is not wrong, just not fully distinguished.
Comment #5: In table 2, check the name with “a-”, “b-”, “c-” or “d” which should be Greek letter alfa or beta etc. In table 5, please give the meaning of the color.
Response: We appreciate to the reviewer for these corrections and comments, the text has been modified accordingly.
Comment #6: Check the format of references, some of them aren’t normative. The Latin names including plants and microorganism should be italic. Only the first letter of the first word in the title should be capital, while others are lowercase, such as ref. 33, 44, 52, 61, 70, 91, etc. All the first letter in the journals should be capital such as ref. 43, 53, 55, etc.
Response: We extend our thanks to the referee for the comment. As requested, we have made the necessary changes by italicizing the text whenever referring to a scientific name and we corrected the capital/lowercase in the references.
Round 2
Reviewer 2 Report
Comments and Suggestions for Authors
Introduction: Line 52 Genista tridentata (L.) Willk. or Pterospartum tridentatum (L.) ......
It will be better if the authors replace or by synonym between parenthesis
Table 1 : Sampling period: NA if the information not available
Table 1 and other table if the Phytochemical (s) not detected please use ND or NR: Not detected or not reported
In our opinion at least one figure which summarizes the important chemical families of reported Phytochemicals in the plant species should be included in the manuscript
Author Response
- Summary
We would like to thank very much for the reviewer’s effort and dedication in evaluating our manuscript. Please find the detailed responses below and the corresponding revisions/corrections highlighted in the re-submitted files. The points addressed by the reviewer allowed us to considerably improve our work.
- General Evaluation
|
Questions for General Evaluation |
Reviewer’s Evaluation |
Response and Revisions |
|
Is the work a significant contribution to the field? |
We appreciate the evaluation. Parte superior do formulárioParte inferior do formulárioThank you |
|
|
Is the work well organized and comprehensively described? |
||
|
Is the work scientifically sound and not misleading? |
||
|
Are there appropriate and adequate references to related and previous work? |
||
|
Is the English used correct and readable? |
- Point-by-point response to Comments and Suggestions for Authors
Comment #1:. Introduction: Line 52 Genista tridentata (L.) Willk. or Pterospartum tridentatum (L.) ...... It will be better if the authors replace or by synonym between parentheses.
Response: In accordance with the reviewer’s suggestion, we have rewrite the sentence and now read:
Lines 56-65: " Genista tridentata (L.) Willk. (the recognized name for this species, also known as Pterospartum tridentatum (L.) Willk., the commonly used name in both scientific literature and commercially available extracts.”
Comment #2: Table 1: Sampling period: NA if the information not available; Table 1 and other table if the Phytochemical (s) not detected please use ND or NR: Not detected or not reported.
Response: We appreciate to the reviewer for these corrections, the text in the table has been modified accordingly. We have incorporated identical changes into tables 2 and 3 to standardize the format of data presentation.
Comment #3: In our opinion at least one figure which summarizes the important chemical families of reported Phytochemicals in the plant species should be included in the manuscript.
Response: In accordance with the reviewer's suggestion, we have included an image displaying the chemical families of the reported phytochemicals (Figure 2). We have also color-coded the table to correspond with each family.
Reviewer 3 Report
Comments and Suggestions for Authors
The manuscript (biology-2649924 - Revised Version) has been revised according to the editor and referees’ comments. I have one comment as follows.
Please check the percent (%) in “with gastric (IC50= 105.8±5.5 %) and pancreatic juices (IC50= 103.5±17.3 %)” in line 325, if it is mm/ml.
Author Response
- Summary
We would like to thank very much for the reviewer’s effort and dedication in evaluating our manuscript. Please find the detailed responses below and the corresponding revisions/corrections highlighted in the re-submitted files. The points addressed by the reviewer allowed us to considerably improve our work.
- General Evaluation
|
Questions for General Evaluation |
Reviewer’s Evaluation |
Response and Revisions |
|
Is the work a significant contribution to the field? |
We appreciate the evaluation. Parte superior do formulárioParte inferior do formulárioThank you. |
|
|
Is the work well organized and comprehensively described? |
||
|
Is the work scientifically sound and not misleading? |
||
|
Are there appropriate and adequate references to related and previous work? |
||
|
Is the English used correct and readable? |
- Point-by-point response to Comments and Suggestions for Authors
Comment #1: Please check the percent (%) in “with gastric (IC50= 105.8±5.5 %) and pancreatic juices (IC50= 103.5±17.3 %)” in line 325, if it is mm/ml.
Response: We would like to thank the reviewer for their persistence in this comment. There is, in fact, an error in our revision. While the values and percentages are accurate, they are not related to the IC50 but rather correspond to the percentage of the activity after digestion. Therefore, in accordance with the reviewer’s suggestion, we altered the text of the manuscript which now reads: “The results showed IC50 values for AChE activity of 1090±4µg/ml. After 4h digestion the remaining enzymatic activities for gastric and pancreatic juices were 105.8±5.5 % and 103.5±17.3 %, respectively.”